# 'The wife should do as her husband advises': Understanding factors influencing contraceptive use decision making among married Pakistani couples—Qualitative study

**Mariyam Sarfraz**[1]*, **Saima Hamid**[2], **Asli Kulane**[3], **Rohan Jayasuriya**[4]

**1** Health Services Academy, Islamabad, Pakistan, **2** Fatima Jinnah Women University, Rawalpindi, Pakistan, **3** Karolinska Institutet, Solna, Sweden, **4** University of New South Wales, Sydney, Australia

* msarfraz@hsa.edu.pk

**Data Availability Statement:** All transcripts and relates files are available at the following: https://www.kaggle.com/datasets/mariyamsarfraz/contraceptive-use-in-couples.

## Abstract

This paper explores the perceptions and attitudes of married couples which prevent them from using modern contraceptive for purpose of family planning, based on semi-structured interviews with 16 married couples from rural Pakistan. This study, with married couples, not using any modern contraceptives, discussed issues of spousal communication and religious norms using qualitative methods. Despite near universal knowledge of modern contraceptives among married Pakistani women, the use continues to be low, with high unmet need. Understanding the couple context about reproductive decision making, pregnancy and family planning intentions is imperative to helping individuals fulfil their reproductive desires. Married couples may have varying intentions and desires about family size; a lack of alignment between partners may lead to unintended pregnancies and affect uptake and use of contraception. This study specifically explored the factors which prevent married couples from using LARCs for family planning, despite their availability, at affordable prices in the study area of rural Islamabad, Pakistan. Findings show differences between concordant and discordant couples regarding desired family size, contraceptive communication and influence of religious beliefs. Understanding the role that male partners play in family planning and use of contraceptives is important in preventing unintended pregnancies and improving service delivery programmes. This study also helped identify the challenges married couples, particularly men have in understanding family planning and contraceptive use. The results also show that while men's participation in family planning decision making is limited, there is also a lack of programs and interventions for Pakistani men. The study findings can support development of appropriate strategies and implementation plans.

## Introduction

Uptake and use of contraceptives for family planning has remained low in Pakistan, despite availability at low cost through a variety of service providers. The low uptake of contraceptives

**Funding:** The author(s) received no specific funding for this work.

**Competing interests:** The authors declare that no competing interests exist.

continues to be explored through academic and programmatic research to identify barriers to use; most common of which are reported to be women's misconceptions about contraceptives, fear of developing side effects and social norms. Early marriage, son preference, husband's desire for more children and their perceived opposition to use are also known to affect Pakistani men and women's use of contraceptives for family planning [1, 2]. Evidence from neighbouring South Asian countries (India, Bangladesh and Nepal) shows that improved spousal communication led to increased concordance among couples on family size, child spacing, while also increasing the uptake of modern contraceptives [3–5]. In the context of Pakistan, as women's use of modern contraceptives, especially use of Long Acting Reversible Contraceptive (LARC) methods, is influenced by the husband/spouse, it is important to understand the couples perspectives regarding family planning and use of LARCs [6–10]. The LARC methods are popular in several parts of the world, however, their use is uncommon among women in Pakistan [11, 12]. Although these LARC methods are provided at subsidised costs through the Pakistan government's family planning and services program, their use is very low (3%) with high discontinuation rates (30% discontinue use within 12-months), especially among young married women [13, 14]. Research shows that women's fear of the intra-uterine devices (IUD) harming the uterus and affecting fertility is the primary reason for not using this LARC method; in addition, experiences of ill-health is the leading cause for discontinuation of use of IUDs [15, 16].

Understanding the couple context about reproductive decision making, pregnancy and family planning intentions is imperative to helping individuals fulfil their reproductive desires. Married couples may have varying intentions and desires about family size; a lack of alignment between partners may lead to unintended pregnancies and affect uptake and use of contraception. The uptake of contraceptives among couples is affected by various factors, which include socioeconomic status, education, religion and wrong perceptions about family planning determine the utilization of modern contraception among Pakistani women [17]. An analysis of the PDHS 2017–18 also established that the perception that contraception is women's business and discussion of family planning with health workers were significant predictors of modern contraceptive usage in Pakistani men [18]. This study attempted to explore the factors which prevent/inhibit married couples using LARCs for family planning, despite their availability, at affordable prices in the study area of rural Islamabad, Pakistan.

## Materials and methods

The data presented in this paper was collected as part of an explanatory mixed methods community-based family planning and social networks research study conducted in rural communities of Islamabad, Pakistan, the goal of which was to understand factors affecting utilization of long-term reversible contraception (LARC) among married couples, in rural areas [19]. The data for this qualitative study is based on in-depth interviews with 16 married couples, purposively identified from the survey respondents, based on their contraceptive use history. Only couples where either the men or the women identified no history of use of LARCs were included in this study. Forty (40) married couples were identified and invited for participation; however, twenty-four (24) couples refused to participate in this research owing to either non-availability of husband at time of data collection (n = 4) and men's refusal to participate (n = 20). In total, sixteen married couples (32 respondents) were interviewed for this study. A written, informed consent was taken from all respondents. Their socio-demographic characteristics, contraceptive use history as well as their family planning intentions are given in Table 1.

**Table 1. Socio-demographic characteristics and contraceptive preferences and use of respondents.**

| Characteristics | Men (n = 16) | Women(n = 16) |
|---|---|---|
| Age | 27 to 41 | 21 to 35 |
| Mean | (35 ± 5 years) | (30 ± 4.6 years) |
| Education | | |
| Illiterate | 3 | 3 |
| Can read/write | - | 1 |
| Primary | 4 | 3 |
| Secondary | 6 | 4 |
| Graduate | 2 | 4 |
| Years married | 4 to 14 years | |
| Range/Mean | (9 years) | |
| Number of pregnancies | 2 to 6 | |
| Mean | (4.4) | |
| Currently pregnant | 1 | |
| No. of children | 2 to 6 | |
| Range/Mean | (3.5) | |
| Previous contraceptive use | Yes = 3 | Yes = 13 |
| | (Condoms (male)– 1; Pills– 1; Injection– 1) | (Condoms (male)– 11; Pills 1; Injection– 1) |
| Desire for more children | Yes = 5 | Yes = 6 |
| | No = 11 | No = 10 |
| Unmet need | Spacing = 5 | Spacing = 5 |
| | Limiting = 11 | Limiting = 10 |
| | Undecided = 0 | Undecided = 1 |

The interview guides developed had a common core of identical questions, with prompts defined for each concept. Respondents were specifically asked about their perspectives on family planning, birth spacing and family size. The interview explored respondents' reasons for not using a contraceptive method. Interviews were conducted in the respondents' homes, in the local language (Punjabi), translated and transcribed in English, for analysis. Interviews with spouses were conducted simultaneously, by same sex interviewers, in different rooms, to prevent interactions between spouses that could skew the information and responses shared. Interviews with women were conducted by the lead researcher (medical doctor with training in public health and qualitative research) and a female research assistant (a qualified nurse with a post-graduation in public health) having considerable experience working with women in rural communities. A male research assistant, qualified health professional with post-graduation in public health, was trained to interview the male respondents, to assist data collection with men.

Data from the interview transcripts and field notes were merged to describe respondent's experiences with use of LARCs, providers' approach to promoting use, as well as, to assist in developing an understanding of low uptake of LARCs, despite its low cost and easy availability A dyadic thematic analysis approach was used to highlight the uniqueness of this data; the results give the perspective of both the male and the female in a dyad [20]; where required, quotes from the couple are given together. A coding structure was developed through mutual discussions within the research team and finalized according to emergent themes, The finalized coding structure was used to code all the transcripts in NVivo 12.

The prolonged engagement of the research team members with the local community and study participants helped establish credibility for the findings [21]. During data collection,

member checking with respondents was employed to validate the researcher's understanding of interview responses. Validity of findings was ensured by undertaking regular debriefing meetings with research team to review information collected, discuss findings and highlight interesting leads to follow for further data collection [21]. Ethical approval for the study was obtained from Human Research Ethics Committee (HREC) of University of New South Wales (UNSW) and National Bioethics Board (NBB) of Pakistan.

## Results

The results reported are based on thirty-two interviews (16 couples). The characteristics of the couples are in Table 1 below. Average age of men was 35 years, while that of women was 30 years. Respondents had mixed levels of education, with most having primary and secondary level schooling. Majority of the women interviewed had a parity of two or more children. At the time of data collection, one couple was expecting their third child.

The results show an unmet need for contraception among the couples interviewed, for either spacing the next pregnancy or limiting future pregnancies. This distinction between the respondent couples was made based on their future childbearing preferences. The definition of unmet need used here is the discrepancy between women's desire to limit further pregnancies while not using any modern contraceptive method [13]. Among the 16 couples interviewed, five couples had an unmet need for spacing childbearing, while the rest did not want any more children. The analysis identified the following three themes: Deficient knowledge and misperceptions about modern contraceptives, Influences on decision-making regarding contraceptive use and Discordance in fertility desires. The themes and sub-themes are given in Table 2 below.

### 1. Gaps in knowledge and misperceptions about modern contraceptives

This theme describes the lack of functional sexual and reproductive health knowledge among both men and women, which led to non-use of modern contraceptives. Respondents also had concerns about side effects of modern contraceptives, which they considered harmful for health of women or children.

**1.1 Lack of functional sexual and reproductive health knowledge.** Functional, sexual and reproductive health literacy (SRHL), defined as knowledge of issues and skills to access, understand, appraise and apply relative information to manage SRH related issues, including contraceptive use for family planning [22], was found to be lacking in the respondents. The results show that respondents had manifold gaps in their knowledge regarding modern contraceptive for instance, they knew about various methods but lacked information on their proper use to avoid pregnancy. Women were more familiar with variety of modern

Table 2. Result themes and sub-themes.

| Themes | Sub-Themes |
| --- | --- |
| Gaps in knowledge and misperceptions about modern contraceptives | Lack of functional sexual and reproductive health knowledge |
| | Misperceptions about side effects |
| Influences on contraceptive use decisions | Influence of husband and family members on fertility decisions |
| | Influence of religious beliefs on contraceptive use |
| Discordance in fertility desires | Limited Inter-spousal communications on family planning |
| | Reproductive (life) events that influence concordance |

contraceptive methods as compared to men. This difference in their level of information could be due to their frequent interactions with local Lady Health Workers (LHWs); the men lacked similar information sources.

### Couple 6

*Wife: I have heard about them (IUD and implant) but I do not know much detail about it. I had no need to use them, you see. If I decide to use it then I would ask Baji (LHW) about it or the doctor in centre here can tell me where to go for them*

The male respondents knew names of different modern contraceptive methods; however, they had limited knowledge about their appropriate use. Men identified their wife as their main source of information about contraceptives; the other sources of information identified included print or electronic media and health care providers, mostly doctors.

### Couple 2

*Husband: I know about these methods, like pills and injections and there is operation also; my wife told me about these. She (wife) found out about these (contraceptive) methods from her sisters in the village and this Baji (LHW), who comes to give polio drops.*

*Wife: I talked to the LHW here and she told me about challah (IUD), Capsule (Implant), injections and tablets. She said to talk to your husband to decide what method you want to use.*

The male respondents highlighted a lack of family planning information sources and platforms for men. Most of the male respondents were of the view that a community-based male information provider, like the LHW, would be useful for giving information on family planning methods. On probing for appropriate site for such a service, men explained that this resource should be at community level, nearer to home as they would not be able to leave during work hours to access such services.

### Couple 11

*Husband: See, not everyone knows how to use internet or go to a website. But if there is a centre for men, we can go there and talk to someone there. Like there is the lady health worker, women talk to them and ask them questions. Same should be there for men, because you see, they are trained to give information and if we don't understand, we can also ask them to explain it properly.*

### Couple 6

*Husband: . . . there should be some proper place for this purpose. At present we don't have such kind of information source for men. I think a centre should be established near our houses or there should be someone at the hospitals to give this information*

The results also show that men and women had a different perspective about benefits of practicing family planning; women viewed use of contraceptives for birth spacing as being beneficial for good health of mother and children, while the men considered family planning, for birth spacing, from the financial perspective of providing for a family.

### Couple 11

*Husband: I believe men have a greater responsibility in this decision as he has to earn money for his family and take care of their needs. It would be very difficult for him to feed his family if children are born without any gap, his earnings will not be sufficient to meet their expenses. I believe it is better to have longer gap between children and do proper family planning.*

*Wife: See if you use some family planning method, then there is a healthy gap between children due to this and each child gets proper attention. A child should have mother's milk for two years, it is his right. If another child is born early, then the first child deprived of this right. Secondly, there are also problems for the mother's health as well.*

**1.2 Misperceptions about side effects.**   Women had several misperceptions about side effects of and ill health due to modern contraceptives; this was cited as the most common reason for not using any method for family planning. The respondents' fear of developing side effects was found to be mostly based on anecdotal information, from social interactions with family and friends.

*Couple 2*

*Husband: People talk a lot about them and say they are not good to use. I have heard they have side effects which are not good for health. She (wife) also said that these things cause infection after you use them, so she did not want to use them.*

*Wife: Some people say Challah (IUD) is good and some say it's not. Injections also have side-effects. Everyone has their own opinion. Women who have used it, they say they had infection like pus formation and pain. I want to avoid that, but I don't really know much about it*

The respondents who had an unmet need of family planning, for spacing the next pregnancy, lacked information about appropriate contraceptives to use; they expressed fear regarding ill-effects of modern contraceptive methods, especially the LARCs, on fertility and ability to conceive a child in future. Thus, instead of using modern contraceptives, the women discussed relying on breastfeeding, post-partum amenorrhea, abstinence or withdrawal methods to avoid an unplanned pregnancy.

The women with an unmet need for limiting further pregnancies were more concerned about the effects of LARCs on menstrual bleeding patterns and uterine infections associated with IUDs. The couples who did not want more children were contemplating use of either a LARC method or an operation (tubal ligation/vasectomy).

*Couple 4*

*Husband: I know about all the contraceptives like there pills and Saathi (condom) method, others are like injectable, and then there is the tube or women also call it challah (IUD); But I think operation is the best solution for this*

*Interviewer: Why is that?*

*Husband: All these methods have a lot of effects on women's health and then they feel weakness, pain and need treatment.*

*Wife: We don't want to have any more children, but I am afraid of using the injection or the challah (IUD), because so many women have bad side effects with it, like they say they have a lot of (menstrual) bleeding, or they start to have pain in the back, all the time. I want to use something that will not give me any problems.*

## 2. Influences on contraceptive use decisions

The results show that the respondents' decision to use modern contraceptives was strongly influenced by the spouse, older family members and their religious beliefs. Notably, all respondents talked about fertility and family size when asked about use of contraceptives for family planning.

**2.1 Influence of husband and family members on fertility decisions.** The men and women interviewed had different opinions about primary decision maker and individuals who can influence a couple on their issues of family planning, contraceptive selection, family size and child spacing. Most of the male and some of the female respondents believed that a joint decision-making approach is the appropriate way for a couple to take decisions on family planning, contraceptive selection, family size and child spacing. However, both men and women also emphasised that husband's agreement for use of modern contraceptives is important and his permission was considered necessary. Most of the women also opined that the main decision maker for contraceptive use and birth spacing is the husband and a wife should comply to his wish. The responses of study participants reflect an influence of local socio-cultural norms pertaining to conduct of life as a married person and related responsibilities.

*Couple 4*

*Husband: Decision about the number of children should be made by both the husband and wife, like they should decide together; but the wife should do as her husband advises*

*Wife: In my opinion, the main role is of the husband. Because if he does not want to use something or he thinks it is not right for the wife to use contraceptive, then how can she use it?*

On probing women about their intentions to use contraceptive methods in future, especially LARCs, women stated having the support of their husbands in starting use of LARCs, if they decided to use it; responses of most of the men interviewed matched this claim. The men confirmed that their wife can choose any contraceptive method to use and were willing to provide the support needed, including taking them to the service provider, bearing cost of the contraceptive services and for treatment, in case they experienced side effects.

*Couple 8:*

*Husband: She (wife) told me about these family planning methods, and I told her to do as she thinks is good for her and it will also be good if she starts using it*

*Interviewer: Right; so, if your wife has decided to use the capsules (Implant) method, you will support her?*

*Husband: Yes; I will take her to centre myself.*

*Interviewer: And if she experiences any side effects, what will you tell her?*

*Husband: I will take her to the doctor to get medicine for it. . .*

*Wife: My husband never told me not to use any family planning method; he never said anything like that. He says I should do what I can easily do. . .*

In terms of role of other family members, respondents identified the influence of their mothers and mother-in-law, on decisions regarding family size. The male respondents acknowledged that although the decision for use of contraceptives should rest entirely with the couple, without interference from any other family member, they recognised the influence of

their mothers on the family size/number of children. Some of the women also identified the influence their mother-in-law had on their family size, gender mix and spacing between the children and use of contraceptives.

*Couple 12*

*Husband: I think husband and wife should decide about the number of children they want and whether they want to use contraception or not. But mother-in-law also has the influence over this decision, like the number of children; they can talk to the daughter-in-law for this*

*Wife: . . . the thing is that my husband was the only son of his parents, so it was my mother-in-law's wish that I have at least two sons. And I think my husband also wanted this*

The results also show that some women had the support of their mother-in-law for use of contraceptives for family planning.

*Couple 14*

*Wife: We do not want any more children . . . my mother-in-law has also asked me to use family planning; she asked me to talk to Baji (LHW) after the birth of my third daughter.*

**2.2 Influence of religious beliefs on contraceptive use.** The analysis revealed that religious and social norms played a significant role in shaping family planning decisions made by most women; respondents used religion driven arguments to justify their non-use of contraceptive methods. On the one hand, certain respondents staunchly believed that family planning violates the fundamental edicts of their faith. On the other hand, many women justified their use of modern contraceptive methods to achieve religion-ordained responsibilities which include maintaining well-being of their own selves, family members and children.

*Couple 6:*

*Husband: The number of children should be left to Allah. We should not intervene with what his decision are for us. See, He brought us to this world and whoever is destined to be born, will be born. We should try to have a space between children but if there is an unplanned pregnancy, that should not be ended.*

*Wife: I think if Allah wants, He will bring a soul to life, to this world. But a person should also plan about this, like how many children to have and when to have them. My two elder sisters do not have kids. But Allah blessed me with 5 children*

Women shared the religious beliefs of their spouse and considered use of contraceptives for family planning as an un-Islamic act. However, simultaneously women also discussed their fears of developing side effects, which they had learned about from their social networks. Moreover, many women considered the traditional (withdrawal/coitus interruptus; called colloquially as 'Islamic method') method a safe choice for spacing pregnancies as it is advised by their religious doctrine and, hence, cannot have poor consequences for individuals. The women's justification for the 'Islamic method' may have been a result of cognitive dissonance due to mismatch between their religious belief concerning practice of family planning, and their desire to prevent an unplanned pregnancy. As a way of addressing this dissonance, the respondent resorted to using traditional methods, which was also supported by their religious belief.

*Couple 15*

*Wife: Our religion says that it is right of the child to be breast fed for two years and it also says that we should not over burden our self beyond our capacity; so, I think family planning is allowed for this purpose. We don't want to have any more children, but I am afraid of using the injection or the challah (IUD), because so many women have bad side effects with it. I prefer the Islami method, because there is nothing which is wrong in our religion.*

Most couples who were interviewed held conservative views about contraception, influenced by their religious teachings; generally, they shared the mindset that it is not permissible. However, the results also show that the couples with unmet need for limiting also used religious belief-based arguments to justify family planning and use of contraceptives. This group of couples considered the concept of family planning to be supportive of religion ordained responsible behaviour for married couples and parents. According to them, Islamic way of life expects that married men ensure welfare of their family, as provider and head of the family; while women are advised to breastfeed their child for two years and ensure good health of the new-born child. As an extension of this, these respondents considered family planning to be useful for general well-being of their family, while also supporting their efforts to fulfil religious obligations (such as providing good food, lifestyle and education to children).

*Couple 12*

*Husband: It is not useless; it is good to plan for such things. See having children or not is in Allah's hands . . .*

*Interviewer: So, will you or your wife start use of a contraceptive method now*

*Husband: I think the Islami method is best for this and we will use that.*

*Wife: He (husband) agrees with me about using family planning, but we have never used anything. He said we can have gap using the traditional/Islamic method (withdrawal).*

## 3. Discordance in fertility desires

Exploration of the data shows that there was discordance about desired fertility and family size between the couples who had an unmet need for spacing. Although the respondents stated knowing their partners' desired family size, their responses about their planned family size were conflicting; there was also mismatch in responses for desired fertility and spacing between children. The couples who were interviewed either made guesses about their partner's desired number of children or had not discussed the issue with their partner.

*Couple 5*

*Husband: Yes, we have talked about this; she said she wants to have four children*

*Wife: I said to my husband three are enough for us, one son and two daughters.*

**3.1 Limited inter-spousal communication on family planning.** Respondents reported having infrequent conversation with their partner on issues of family planning and described difficulties with communicating about contraceptive use. Although women were more knowledgeable and aware of different contraceptive methods, they admitted their shyness in openly discussing such issues with their husbands.

*Couple 7*

*Husband: No, me and my wife have never talked about it as I think I don't have enough knowledge about what we can use for family planning*

*Wife: I don't know how to talk to him; I asked him about using the pills, but he said to not use them because I am breastfeeding our son and the pills have bad effects on that.*

*Interviewer: How do pills affect breastfeeding?*

*Wife: He said it reduces the milk supply and I will not be able to feed the baby*

Some of the respondents had not talked about their on desired family size with their spouses. In exploring the reasons for this, the male respondents highlighted their insufficient knowledge about family planning methods as a reason for not discussing such issues with their spouses. The women, on the other hand, had shared contraception related information with their husbands, but did not talk about their preferred contraceptive. Moreover, the women also appeared to be unsure about effectiveness of contraceptives, especially the LARCs.

*Couple 10*

*Husband: . . . it is because of lack of knowledge and information about the contraceptive methods and family planning which I think is the main reason we are not talking about it or using anything. And secondly, I would say shyness is among the main factors.*

*I: Shyness about what?*

*Husband: Shyness in talking about contraceptives, with anyone, including my wife. I think if someone, like a doctor, tells me this by explaining everything and if I'm able to have a conversation about it, then I think I can overcome the shyness about this*

*Couple 15*

*Wife: No, no, I never talked to him on this matter. . . . . .*

*I: Any reason for not talking about this with him?*

*Wife: Hmm . . . .. no particular reason. I just ask my sister about these things. I'm not comfortable in talking about family planning with my husband; he is very shy about that. He does not like to talk about it.*

The results also indicate at the disparity existing among couples related to use of contraception and its benefits for them. The results imply that couples' lack of communication was perhaps not new in their relationship; limited conversations about contraception and pregnancy led to unintended pregnancy, irregular and covert use of contraceptives. Almost all the couples interviewed were not using any contraceptives, despite clear desires to avoid pregnancy. Most women shared frequent use of condoms by their husband, for preventing unwanted pregnancies; comparatively, only one male respondent accepted the use of condoms. In some cases, women also reported clandestine use of a hormonal contraceptive (pills or injections), as their husband was not in favour of using modern contraceptives for family planning.

*Couple 6*

*Husband: No, did not use anything for family planning but now my wife wants to get operation done or that capsule (implant)*

*Wife: I started using injections, without telling my husband about it because he was not in favour of it*

The contradictory statements demonstrate a lack of effective communication regarding use of contraceptives between couples. Similar gaps in communication were found for other issues related to family planning and child spacing. Responses from most of the men showed that they favoured use of modern contraceptives and would support their wife's decision to use them. However, women's responses contradicted these claims where they narrated a lack of support or opposition from their husbands regarding contraception.

*Couple 3*

*Husband: The lady health worker here, she used to come to my wife regularly and she told her about these methods, but we did not use anything before*

*Wife: I started using injections after my fourth child, without telling my husband about it. . . . . . . . . But after our youngest daughter was born, I told him, and he agreed to my using them.*

*Interviewer: Why did you not tell him about using injections?*

*Wife: He was not in favour of injections initially; he used to say as Allah is blessing us with children so we should not do anything to stop it. But after the birth of my last child, he accepted this thing.*

**3.2 Reproductive (life) events that influence concordance.** Results of the study show that two factors influenced concordance between the couples on decision to use LARC methods: desired family size and unplanned reproductive events. Among all the couples interviewed, interspousal communication on family planning and size was at most times initiated either after closely spaced birth of two or three children or an unplanned pregnancy

*Couple 8*

*Husband: Yes, we have talked about the number of children we should have; three are enough for us, we have a son and two daughters*

*Wife: After my youngest daughter was born, I said to my husband these are enough, one son and two daughters. But I think my husband wants another son.*

*Interviewer: Did you talk to him about this, the number of children you want?*

*Wife: No; we never talk about this*

At time of data collection, some of the couples who had an unmet need for spacing, were contemplating use of contraceptive methods for family planning. Although their responses were discordant, both the husband and wife were considering initiating the use of modern contraceptive methods to avoid future unintended pregnancies; however, such view were limited in number and were not shared by all respondents.

*Couple 6*

*Husband: I talked to her about this (contraceptive use) and told her to use those pills or injections she was telling me about. She (wife) said that both of these things cause infection after you use them, so she did not want to use them.*

*Wife: I told him (husband) about family planning methods I heard about at the camp; like I told him about pills and injections and the challah (IUD) also. Baji (LHW) told me that challah (IUD) is good to use for two to three years, so maybe I will get that. I have not decided yet . . .*

On the other hand, some of the couples who considered to have completed their desired family size, with an unmet need for limiting childbearing, had concordant views about their desired family size. The data shows that this group of couples decided to use a LARC method after experiencing an unplanned or mistimed pregnancy. Among the couples interviewed, few also shared their experience of attempting to terminate their unwanted pregnancy, as a joint decision. However, some of the men also reported asking their wives to continue with the pregnancy, considering it to be God's will.

<u>Couple 2</u>

*Husband: When my wife told me about the last pregnancy, I didn't want that child but then my wife tried to end that pregnancy and it did not work, so then I decided that we should have that baby and use something after her birth. My wife wants to get the operation (tubal ligation) done.*

*Wife: . . . I will tell you honestly that when I had my last pregnancy, I tried to terminate it with different methods like I took tablets and did jumping, used home remedies and drank herbal teas, but nothing happened. I told my husband and he said that we should accept this, and Allah blessed me with a daughter.*

## Discussion

In the context of low uptake of LARC in Pakistan, this study sought to explore the perceptions, motivations, and social influences on inter-spousal decision-making process, of those married couples not using LARC as a contraceptive method. As women's use of LARCs is influenced by their husband, we studied couples, a design although used in other countries, but not used much in Pakistan [4, 23]. The following three themes emerged from the qualitative data: i) Gaps in knowledge and misperceptions about modern contraceptives; ii) Influences on contraceptive use decision and iii) Discordance in fertility desires.

### Deficient sexual and reproductive health literacy

The findings of the study revealed that men and women have insufficient sexual and reproductive health literacy (SRHL) [22] regarding use of modern contraceptives; moreover, they feared suffering its side effects. Inadequate SRHL is linked to higher incidence of adverse sexual and reproductive outcomes, sexually transmitted infections and unintended pregnancies [22, 24]. The national surveys conducted to ascertain FP knowledge and practice in Pakistan such as the Pakistan Demographic Health Survey (PDHS) report near universal knowledge (98%) of modern contraceptives. However, these survey only measures respondents' awareness of names of modern methods and does not ascertain aspects of access, appropriateness, and procedures to use the methods. Despite the reported high levels of knowledge, the modern contraceptive prevalence rate (CPR) is 25% to 26% as per the past three PDHS conducted in Pakistan [14, 25, 26]. Our qualitative study revealed that both women and their spouses did not have a functional understanding of reproductive health and contraceptive use. This may be due to two reasons: low education levels among residents of rural areas and limited

information provided to them, which was tailored to their education level at these settings. Research among Pakistani women have shown a positive association of educational level with modern contraceptive use [27, 28].

In terms of the information sources for married men and women included in this study, the LHWs and electronic/print media were identified as the most common source of information; like findings reported in the latest PDHS [26]. The study also found that women shared information given by the Lady Health Workers (LHWs) with their husbands. In the cultural context of Pakistan, the LHWs interact almost exclusively with women on issues of reproductive health. The study found that men felt the absence of a similar source for information to be a gap. The Population Welfare Department (PWD) attempted to address men's family planning information and services needs by appointing a cadre of male social mobilizers in early 2000s, however, this service has remained largely ineffective [29, 30].

Importantly, results of our study found that LHWs have gaps in their knowledge of contraceptives and their use, based on reports by women they counselled. The evidence evaluating services of this cadre of service providers shows that they lacked appropriate training and organizational support to provide necessary services [31]. This may be due to the LHWs engagement with several other public health activities which consumes the allocated time for discussion with women on issues related to family planning [31, 32]. Considering the results, there is a need to develop and introduce alternative interventions which comprehensively address SRH and family planning knowledge deficits among men and women [33–35].

## Influences on contraceptive use decisions

**Influence of husband and family members on fertility decisions.** Results show that women communicated mostly with older family members or local health workers on family planning issues. As prior research has demonstrated links between counselling by an older female and improved contraceptive uptake in similar settings [35–38], the respondents in this study also mostly sought support from the mother or older female siblings for use of contraception. In the population studied, keeping up with the cultural context (patrilineal), the mother-in-law also had an influence on couples' fertility decision. The study found that most men and some women respondents believed that a joint decision-making approach is the appropriate way for a couple to take decisions on family planning, contraceptive selection, family size and child spacing. However, the requirement for husband's agreement for use of modern contraceptives was emphasized and his permission was considered necessary, especially for LARC use decisions. With reference to women's intentions to use contraceptive methods in future, especially LARCs, women stated having the support of their husbands was important and most spouses confirmed that their wife can choose the LARC method to use and were willing to provide the support needed. The men have a key role in decision making for uptake of LARCs, however, the design of service provision which is directed towards women. Contrary to the culture, this aspect of services assumes that fertility related issues are women's responsibility who have autonomy in decision making for contraceptive use, particularly the LARCs [9]. It is also important to consider here the interplay of gender roles, women's autonomy and contraceptives use decision making among Pakistani couples. Evaluation of the PDHS data sets show that women's contraceptive use is influenced by the men's decision making role. In a patriarchal society like Pakistan, the gender norms put men as financial providers, marking them the decision makers, while women have relatively limited empowerment in matters related to family size, timing and spacing of children and the use of contraceptives. The influence of gender norms on uptake of contraceptives is more pronounced in rural households, as the results of this study show.

The important role of men in developing intentions for child spacing, family planning and decision making for contraceptive use is largely ignored, indicating the need for multidimensional, gender transformative programs. While much attention is focused on role of gender and power imbalance between men and women, it is also important to understand the construct of masculinity practiced in the Pakistani culture. Men are expected to be financial providers, have large families, make key decisions for their wife and children and let older women decide about timing and spacing of children. These constructs maybe masking/affecting men's need for information about contraceptive use, family planning and reproductive health in general. The results of this study clearly indicate the men's desire for access to information, however, Pakistan's family planning program portrays men as 'supportive partners', rather than 'active users' of contraceptive services.

**Influence of religious beliefs on contraceptive use.**   There is limited research exploring the effects of Islamic religious ethos on individual beliefs and behaviours, contributing to cognitive dissonance for using modern contraceptives for family planning. The evidence exploring the effects of religious beliefs and behaviours, contributing to explanation for use of modern contraceptives for family planning among Muslim populations shows that higher degree of religiosity is associated with negative attitudes for nonprocreative sexual interactions [39–42]. Findings from this study show that the attitude and behaviour of couples towards family planning and contraceptive use was influenced by prevalent religious beliefs.

Our study found that FP decisions made by most women to use traditional contraceptive methods of withdrawal (called "Islamic method") and abstinence were to conform to religious injunctions. Some respondents staunchly believed in the notion that family planning violates the fundamental edicts of their faith. On the other hand, many women and men justified their use of planning methods to achieve religion-ordained responsibilities such as to maintain well-being of their own selves, family members and children by limiting the family to what they can afford. Research exploring use of family planning in norther regions of Pakistan, among a religious minority, shows that women used moderate interpretations of Islam which support smaller families to legitimize the use of modern contraceptives

Past research in Pakistan focused on identifying religion related barriers and misconceptions to contraceptive use, identified religious concerns as key reason for not using contraceptives [25]. To address the issue, Population Welfare Department developed behaviour change communication strategy which highlighted religious support for practicing family planning which was somewhat similar to the strategy of the government of Bangladesh [43]. The intervention was piloted by mobilizing religious leaders, local clerics, health care providers and community-based health workers to engage with local communities on the issue of religious barriers to contraceptive use [29, 44]. However, the intervention was not scaled up for wider, sustained diffusion; and hence its intended effect of addressing the religion and contraception debate at a larger scale remains to be explored.

**Interspousal communication on family planning and contraceptive use.**   Interspousal communication is shown to be positively related to an improved uptake and continued use of contraceptives by married couples in similar context [5, 9, 45]. The results of this study show that due to absence of interspousal communication about family planning, respondents did not know their spouse's preference for family size and contraceptive use. The respondents highlighted lack of knowledge about contraceptives and personal inhibitions as factors affecting inter-spousal communication. The reproductive and family planning intentions of the couples interviewed for this study did not align, for which lack of communication on the issue emerged as an important factor. The inhibitions maybe associated with the culturally grounded social norms/practices which prohibit discussion of issues related to reproductive health [46].

The lack of interspousal communication on contraceptive use among married couples not using LARC and men's reluctance to participate in this study can be explained by several factors. The study found instances where respondents had incorrect perceptions of their spouse's intentions about family planning; men and women had misperceptions and did not have adequate knowledge and information about contraceptives. Evidence shows that spousal communication and contraceptive use is low in settings where reproductive decision making is influenced by other relatives, especially mother-in-law [36, 37, 47, 48], as was reported in the results of this study. This supports the finding of women in this study having more children than they wanted owing to the wishes of the husband or mother-in-law. The family planning service providers in Pakistan encourage women to discuss their contraception preferences with their spouses and to convince them to use contraceptive methods for their well-being. However, the study results revealed that couples' deficient knowledge about contraceptives kept them from engaging in meaningful conversations on family planning and contraceptive use. These findings further reiterate the need for improving men and women's knowledge of contraceptives and development of service provision platforms for men.

Interspousal communication is known to be central to contraceptive decision making and use, increased positive communication between the married couples could help reduce unintended pregnancies among married couples [4, 5, 49, 50]. In settings where communication on sensitive issues such as family planning were uncommon, behaviour change campaigns through the mass media have prompted couples to communicate with each other on the use of contraceptives [3, 51]. Evidence shows that women are likely to initiate discussion on contraceptive use in cases where they consider their spouse to be more supportive of women's decisions [4, 36, 47, 52]. Women are also shown to talk about contraceptive use with their spouse when they want to have fewer children, regardless of exposure to behaviour change messages about family planning [53]. Our findings suggest that family planning programs and service providers can play an important role in promoting positive communication between married couples. Service providers can help promote and normalize communication between married couples during family planning visits and discuss concerns regarding joint decision making. Normalizing intimate communications between married couples could increase their confidence in discussing the stigmatized and taboo issue of contraceptive use. Another aspect to be cognizant of is that the Pakistani culture is relatively conservative, such that issues of reproductive health and family planning are considered taboo and discussion on these among men is uncommon [8, 35, 54]. Secondly, the family planning services in Pakistan are women oriented; majority of the community- and facility-based service providers are female. Obstetric-care and family planning services are provided almost exclusively by female providers, in facilities which provide a very limited access to men. This feature of service provision has affected availability of family planning information resources for men. In terms of family planning information sources for men, there is a cadre of male community workers, called Male Mobilizers for family planning awareness raising among men. However, this cadre has never really been functional in increasing male engagement with the family planning services utilization [29]. Apart from the male community workers, there are no other specific family planning services for men; thus, the role of men as partners and decision makers appears to be overlooked by the family planning program of Pakistan. Considering the role men have in decision making for family planning and modern contraceptive use, and the strong evidence supporting their involvement [9, 33, 34, 54], it is important to develop family planning information sources and services for men.

Comprehensive sexuality education programs may be particularly successful in promoting positive communication among married couples and healthy reproductive behaviours early in the life course [55]. Considering the success of peer education and interpersonal

communication initiatives with African men in promoting healthy reproductive and maternal health practices [45, 56, 57], similar initiatives maybe be of immense benefit if contextualized and tested in Pakistan.

The existing gender norms in Pakistan have pinned (use another word) reproductive health, particularly the aspects around pregnancy, child birth, family planning and contraceptive use, as being the sole domain of women, with the older women making decision for the younger. The men are positioned as supportive 'by-standers, and not engaged as partners in decision making. Although there is a growing realisation of involving men in reproductive health, the existing programs and interventions continue to provide services in the traditional manner/through the structure designed for service provision to women and not couples or men.

## Conclusions

This study used qualitative methods to explore the perceptions, motivations, and social influences affecting inter-spousal decision-making process regarding contraceptive use, of those married couples who were not using LARC method for FP. As it is evident from results of analysis that respondents lacked functional knowledge of family planning and modern contraceptives; moreover, they also had misperceptions about side effects of LARCs. The couples' decision to use contraceptives was influenced by individuals as well as reproductive events. The opinion of husband and family members, especially mother-in-law, were reported as being integral to choosing LARC. Experience of successive, short spaced or mistimed pregnancies prompted the couples to initiate discussion on family planning and use of contraceptives. The interspousal communication among the couples also varied; the couples with unmet need for limiting had better communication as compared to the other group. The couples with an unmet need for limiting had concordant views about their desired family size and intentions to use contraceptives; furthermore, the respondents' decision to use contraceptives for family planning was also influenced by religious beliefs. Understanding the role that male partners play in family planning and use of contraceptives is important in preventing unintended pregnancies and improving service delivery programmes. This study also helped identify the challenges married couples, particularly men have in understanding family planning and contraceptive use. The results also show that while men's participation in family planning decision making is limited, there is also a lack of programs and interventions for Pakistani men. The study findings can support development of appropriate strategies and implementation plans. The family planning program managers and policy makers need to consider men as 'active partners' in family planning rather than the supportive role expected of married men. Policy makers and program managers need to consider integrating gender-transformative interventions that challenge existing norms for men's involvement/engagement in family planning and incorporate approaches/actions to target change at personal, community and societal level.

## Author Contributions

**Conceptualization:** Mariyam Sarfraz, Saima Hamid, Rohan Jayasuriya.

**Data curation:** Mariyam Sarfraz.

**Formal analysis:** Mariyam Sarfraz, Saima Hamid, Rohan Jayasuriya.

**Investigation:** Mariyam Sarfraz.

**Methodology:** Mariyam Sarfraz, Saima Hamid.

**Project administration:** Mariyam Sarfraz.

**Software:** Mariyam Sarfraz.

**Supervision:** Mariyam Sarfraz, Saima Hamid, Rohan Jayasuriya.

**Validation:** Mariyam Sarfraz.

**Writing – original draft:** Mariyam Sarfraz.

**Writing – review & editing:** Saima Hamid, Asli Kulane, Rohan Jayasuriya.

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
