## [Decision Letter · Decision Letter 0]

1 Feb 2022

PONE-D-21-32104‘Decision about number of children should be done by both husband and wife; but the wife should do as her husband advises’: Understanding factors influencing contraceptive use decision making among married Pakistani couples – qualitative study.PLOS ONE

Dear Dr. Sarfraz,

Thank you for submitting your manuscript to PLOS ONE. After careful consideration, we feel that it has merit but does not fully meet PLOS ONE’s publication criteria as it currently stands. Therefore, we invite you to submit a revised version of the manuscript that addresses the points raised during the review process.

Two reviewers have provided constructive feedback and suggestions to strengthen the manuscript, including clarification of inclusion criteria (and its implications on findings) and further exploration of gender dynamics.

We look forward to receiving your revised manuscript.

Kind regards,

Hannah Tappis, DrPH, MPH

Academic Editor

PLOS ONE

Journal Requirements:

2. Please provide additional details regarding participant consent. In the ethics statement in the Methods and online submission information, please ensure that you have specified (1) whether consent was informed and (2) what type you obtained (for instance, written or verbal, and if verbal, how it was documented and witnessed). If your study included minors, state whether you obtained consent from parents or guardians. If the need for consent was waived by the ethics committee, please include this information

"Unfunded study"

4. We note that you have stated that you will provide repository information for your data at acceptance. Should your manuscript be accepted for publication, we will hold it until you provide the relevant accession numbers or DOIs necessary to access your data. If you wish to make changes to your Data Availability statement, please describe these changes in your cover letter and we will update your Data Availability statement to reflect the information you provide

Reviewers' comments:

Reviewer's Responses to Questions

**Comments to the Author**

1. Is the manuscript technically sound, and do the data support the conclusions?

Reviewer #1: Yes

Reviewer #2: Yes

2. Has the statistical analysis been performed appropriately and rigorously? 

Reviewer #1: N/A

Reviewer #2: Yes

3. Have the authors made all data underlying the findings in their manuscript fully available?

Reviewer #1: No

Reviewer #2: Yes

4. Is the manuscript presented in an intelligible fashion and written in standard English?

Reviewer #1: Yes

Reviewer #2: Yes

5. Review Comments to the Author

Reviewer #1: Thank you for sharing this interesting and important article. The article is well written and methodologically coherent. The article provides a good overview of some of the key barriers to using LARC as expressed by women and husbands. However, what’s missing is an exploration of the tensions between husbands and wives. Further exploration of these areas where there is a lack of alignment how that translates to practices, method choice, and unmet need would add depth to the article.

I have a few other suggestions/comments to improve the article:

•A more focused analysis of the gender dynamics, a few thoughts for consideration:

o Men’s refusal to participate in study was very high and can lend insights into gender dynamics, and the perception/gender allocation of family planning

o Large discordance between husband wife with respect to previous contraceptive use, specifically with respect to male methods, what does that also teach us about how masculinity is constructed, what insights how confident in measures of unmet need given limited disclosure, might be worth locating analysis in this larger landscape

•Interviews are conducted but husband and wife comments are presented together, please clarify in text

•May be worth mentioning the structural limitations that constrain LARC use in introduction/discussion

Reviewer #2: The paper is well written and provides useful information. There are some minor errors like:

- first sentence of the last para on page 15 is missing a word as its not making sense.

- first sentence on page 26 .....as well has reproductive ...., this should be "as well as reproductive".

I also think that the title is not defining the results very clearly as its not only husbands who influence contraceptive use decision making, but also other family members and religious beliefs. Looking at the results and discussion, it seems the title may highlight the issue of routinely missing men in contraceptive awareness raising and service provision that leaves them too low in knowledge and understanding to be able to support or even guide their wives.

While describing the sampling and inclusion criteria, it is mentioned that only couples not using LARCs were invited for the qualitative study although they were still using short-acting methods. However, the rationale for this selection is not clarified and if similar criteria has been used by other researchers to identify issues around contraceptive use decision making. In addition, it is not mentioned if these couples had any exposure to contraceptive awareness raising programs to determine if their choice of not using LARCs was despite the counseling services or it was due to a weak link in contraceptive counseling programs missing out on husbands and men in general.

6. PLOS authors have the option to publish the peer review history of their article (what does this mean?). If published, this will include your full peer review and any attached files.

Reviewer #1: No

Reviewer #2: **Yes: **Dr Qudsia Uzma

---

## [Author Response · Author response to Decision Letter 0]

31 May 2022

The Response to Reviewers has been uploaded as a separate document.

---

## [Decision Letter · Decision Letter 1]

8 Aug 2022

PONE-D-21-32104R1Decision about number of children should be made by both the husband and wife; but the wife should do as her husband advises’: Understanding factors influencing contraceptive use decision making among married Pakistani couples – qualitative studyPLOS ONE

Dear Dr. Sarfraz,

Thank you for submitting your manuscript to PLOS ONE. After careful consideration, we feel that it has merit but does not fully meet PLOS ONE’s publication criteria as it currently stands. Therefore, we invite you to submit a revised version of the manuscript that addresses the points raised during the review process.

We look forward to receiving your revised manuscript.

Kind regards,

Hannah Tappis, DrPH, MPH

Academic Editor

PLOS ONE

Journal Requirements:

Additional Editor Comments (if provided):

Please disregard comments from the first reviewer regarding not seeing a point-by-point response to comments, but do carefully consider their concern regarding the lack of nuance in discussion about cultural context and gender norms, as well as constructive comments from reviewer #3.  

Reviewers' comments:

Reviewer's Responses to Questions

**Comments to the Author**

1. If the authors have adequately addressed your comments raised in a previous round of review and you feel that this manuscript is now acceptable for publication, you may indicate that here to bypass the “Comments to the Author” section, enter your conflict of interest statement in the “Confidential to Editor” section, and submit your "Accept" recommendation.

Reviewer #1: (No Response)

Reviewer #3: (No Response)

2. Is the manuscript technically sound, and do the data support the conclusions?

Reviewer #1: Yes

Reviewer #3: Yes

3. Has the statistical analysis been performed appropriately and rigorously? 

Reviewer #1: N/A

Reviewer #3: Yes

4. Have the authors made all data underlying the findings in their manuscript fully available?

Reviewer #1: No

Reviewer #3: Yes

5. Is the manuscript presented in an intelligible fashion and written in standard English?

Reviewer #1: Yes

Reviewer #3: Yes

6. Review Comments to the Author

Reviewer #1: Authors did not adequately address comments about lacing in gender through the manuscript, no point by point rebuttal letter was provided so unable to understand why this decision was made. Current discussion about cultural context and gender norms is reductive and tends to essentialize cultural norms without making space for differences/discussing nuance.

Reviewer #3: Hello - Overall, I thought this research was done well and they found interesting results. Below are some comments/suggestions for authors.

1) Please highlight the implications for change based on the results you found (the "so what" question for your findings); I would also include that as part of the abstract - given what you found, what would you suggest changing.

2) Please emphasize what's different about your study and what your findings add to the current literature.

3) Could you provide more details regarding the translation process in your methods; were the transcripts translated into English and then analyzed?

7. PLOS authors have the option to publish the peer review history of their article (what does this mean?). If published, this will include your full peer review and any attached files.

Reviewer #1: No

Reviewer #3: No

---

## [Editor Report · Decision Letter 2]

24 Oct 2022

'The wife should do as her husband advises’: Understanding factors influencing contraceptive use decision making among married Pakistani couples – qualitative study

PONE-D-21-32104R2

Dear Dr. Sarfraz,

We’re pleased to inform you that your manuscript has been judged scientifically suitable for publication and will be formally accepted for publication once it meets all outstanding technical requirements.

Kind regards,

Hannah Tappis, DrPH, MPH

Academic Editor

PLOS ONE

Additional Editor Comments (optional):

Concerns and feedback shared by peer reviewers has been adequately addressed. Minor copyediting is still needed. For example, in the Introduction there are a few sentences with citations in brackets (e.g. "ref MacQuarrie") that do not follow standard references formats used in the remainder of the manuscript.
---

## [Editor Report · Acceptance letter]

16 Nov 2022

PONE-D-21-32104R2 

‘The wife should do as her husband advises’: Understanding factors influencing contraceptive use decision making among married Pakistani couples – qualitative study 

Dear Dr. Sarfraz:

I'm pleased to inform you that your manuscript has been deemed suitable for publication in PLOS ONE. Congratulations! Your manuscript is now with our production department. 

Kind regards, 

on behalf of

Dr. Hannah Tappis 

Academic Editor

PLOS ONE